# Gelatin-Coated Magnetic Nanowires for High-Sensitivity Optical Labels

**DOI:** 10.3390/nano13010015

**Published:** 2022-12-20

**Authors:** M. Charbel Cuevas-Corona, J. Mauricio Lopez-Romero, Alejandro Manzano-Ramírez, Rodrigo Esparza, Rosa E. Zavala-Arce, Alejandro J. Gimenez, Gabriel Luna-Bárcenas

**Affiliations:** 1Center for Research and Advanced Studies of the National Polytechnic Institute, Cinvestav Querétaro, Querétaro 76230, Mexico; 2Centro de Física Aplicada y Tecnología Avanzada, UNAM, Querétaro 76230, Mexico; 3National Technological Institute of Mexico, Tecnológico Nacional de México/Instituto Tecnológico de Toluca, Metepec 52149, Mexico

**Keywords:** encapsulation, detection labels, magnetic nanowires, optical markers

## Abstract

The encapsulation of magnetic nickel nanowires (NiNWs) with gelatin is proposed as an alternative for optical label detection. Magnetic nanowires can be detected at very low concentrations using light-scattering methods. This detection capacity could be helpful in applications such as transducers for molecular and biomolecular sensors; however, potential applications require the attachment of specific binding molecules to the nanowire structure. In the present study, a method is presented which is helpful in coating magnetic nanowires with gelatin, a material with the potential to handle specific decoration and functionalization of the nanowires; in the first case, silver nanoparticles (AgNPs) are efficiently used to decorate the nanowires. Furthermore, it is shown that the synthesized gelatin-coated particles have excellent detectability to the level of 140 pg/mL; this level of detection outperforms more complex techniques such as ICP-OES (~3 ng/mL for Ni) and magnetoresistance sensing (~10 ng/mL for magnetic nanoparticles).

## 1. Introduction

Emerging technologies based on nanomaterials offer solutions to different technological applications. Many of these materials respond to external stimuli and emit optical, electrical, and mechanical signals. Using these materials as labels in immunoassays makes it possible to identify several biological analytes for detecting diseases such as cancer [1] Currently, bio-functionalized magnetic nanoparticles are being used in biodetection because they can be selectively manipulated and attached to biological systems [2,3,4,5,6,7].

Recently, our research group reported a magnetic aligning device capable of detecting magnetic nanowires in a colloidal suspension; this concept is based on the intensity change of scattered laser light due to the alignment change of the magnetic particles when a magnetic field is applied. This method efficiently detects magnetic nanowires down to concentrations of 2 ng/mL suspended nickel nanowires (NiNWs) [8,9].

Functionalization and decoration of the magnetic nanowires are crucial factors in achieving the use of these nanowires to detect specific analytes; we have found that direct covalent attachment of molecules to the surface of NiNWs is challenging, unstable, and sometimes detrimental to their optical detectability. Although more studies must be performed in this direction, in this work, the encapsulation of the nanowire inside a gelatin matrix is proposed that could easily be functionalized via covalent binding or by embedding molecules and nanoparticles.

Gelatin is used as an encapsulating agent and offers many advantages over other polymers due to its abundance, low cost, biological compatibility, physicochemical properties, organic origin, and the presence of carboxyl (-COOH) and amino (-NH2) groups [10]. These groups can interact with metallic surfaces, allowing the gelatin to act as a stabilizing agent in the synthesis of metallic particles that generate stable colloids [11,12,13]. Furthermore, functional groups can bind to different biomolecules, making gelatin an attractive coating material for medical and biosensing applications [10,14].

NiNWs have been used in several applications such as magnetic cell manipulation and the fabrication of sensors [15,16], and the fabrication method of using electrodeposition over porous alumina substrates stands out due to the ease of their synthesis and control of their dimensions [17,18,19]. In addition, NiNWs have magnetic properties and exhibit high light-scattering changes when a magnetic field is applied [20,21].

We propose a two-stage synthesis procedure for the encapsulation process: (1) coating NiNWs with gelatin to provide particle stabilization and biological compatibility, and (2) impregnating the gelatin coating with silver nanoparticles (AgNPs). Furthermore, the addition of AgNPs sets an example, demonstrating how the proposed technique can help decorate the nanowire–gelatin structures with other required artifacts; for instance, the addition of proteins will serve as a basis for bio-detection applications [22].

## 2. Methodology

### 2.1. Ni Nanowires

NiNWs were fabricated via electrodeposition using anodized aluminum as a template in a procedure described by Nielsch et al. [19] with minor changes. First, pure Al foil anodization was performed at 30 VDC in an oxalic acid solution of 0.3 M at 20 °C for 20 min. Next, metal electrodeposition into the porous alumina film was carried out in a saturated solution of NiSO_4_ and H_3_BO_3_ (45 g/L). A nickel counter electrode was the voltage source (6 VAC at 60 Hz). NiNWs were released by dissolving the alumina film in a NaOH 0.3 M solution. Next, NiNWs were washed via ultrasound dispersion followed by precipitation using a neodymium magnet and redispersion in deionized water; this procedure was repeated three times.

### 2.2. Ag Nanoparticles

Gelatin type A (1.25 g) and AgNO_3_ (0.025 g) was dissolved in 25 mL deionized water under stirring and constant heating at 45 °C; after homogenization for 5 min, to trigger the photoreduction of silver ions, the dispersion was exposed to light from an ultraviolet LED lamp (360 nm at 2 W) for 60 min.

### 2.3. Gelatin-Coated NiNWs

The coating of NiNWs was carried out via a two-step desolvation method previously described by Coester et al. [23,24] with minor changes. In short, 1.25 g gelatin was dissolved in 25 mL deionized water under stirring and heating at 45 °C, and 25 mL acetone was added to this solution with stirring for 20 min as a desolvating agent to precipitate the high-molecular-weight gelatin. Next, the supernatant was discarded, and the remaining sediment was solvated again in 25 mL acetic acidic solution (pH = 2.5) under heating; 1 mL NiNW solution (37 µg/mL) was added to the gelatin solution, and it was dispersed via ultrasound for 4 min at 30 watts. Then, the gelatin was dissolved via dropwise addition of 50 mL acetone under heating at 45 °C and non-magnetic stirring. After 10 min of stirring, 500 µL glutaraldehyde (8%) was added to crosslink the particles, and stirring continued for 30 min. Finally, the particles were purified via threefold centrifugation (6000 G for 10 min) and redispersion in acetone/water (30/70 Vol.) using sonication (30 W for 4 min.); the last redispersion was performed only in water. Next, nanowires were separated using a neodymium magnet, and the resultant nanoparticles were stored at 4 °C.

The process described above was also used for coating of NiNWs with silver nanoparticles. In this case, the precursor AgNO_3_ was added to the gelatin solution to form a colloidal dispersion; chemical reduction to produce AgNPs was triggered by UV radiation from an ultraviolet LED lamp (365 nm at 2 W) for 60 min under stirring and heating at 45 °C. The complete process of elaborating NiNWs coated with AgNPs and gelatin is schematically described in Figure 1.

### 2.4. Structural Characterization

NiNW morphology was imaged via scanning electron microscopy (SEM; JEOL model JSM-7610F Field-Emission Scanning Electron Microscope with EBSD, Oxford). Additionally, the morphologies of the NiNWs, gelatin-coated NiNWs, and AgNPs were imaged via scanning/transmission electron microscopy (STEM) using a Hitachi SU8230 Cold Field-Emission Scanning Electron Microscope (CFE-SEM) using the bright-field (BF) and dark-field (DF) detectors; these imaging techniques help determine the presence of NiNWs, AgNPs, and gelatin.

### 2.5. Chemical Characterization

Chemical analysis of the gelatin-coated NiNW colloidal suspension was conducted using an inductively coupled plasma optical emission spectrometer (ICP-OES) HORIBA to determine the nickel quantity in parts per billion (ppb) in all samples.

### 2.6. Detection

Coated nanowires were suspended in deionized water in an optical cuvette, and they were detected with the aid of the experimental device shown in Figure 2; this device has been previously reported by our group [8]. In brief, this device can generate two orthogonal magnetic fields of 100 Gauss created by four copper coils installed on each side of the cuvette. A set of two Helmholtz copper coils helps to induce a magnetic field in the direction of the laser beam (650 nm @ 5 mW), and other sets of two coils aligned perpendicularly will also induce a magnetic field in a perpendicular direction to the laser beam. Synchronizing the magnetic fields on/off in each direction promotes the rotation of the nanowires. This on/off procedure causes a change in the scattered light intensity of the rotating nanowires; this phenomenon is the key to detecting suspended nanowires. After removing the magnetic field, the nanowires lose alignment due to Brownian motion. On top of the measured sample is placed a photodetector sensor that measures changes in the scattered light intensity. Signal changes are electronically amplified using operational amplifiers, digitally converted using a microcontroller, and then, the digital data are transferred to a PC for filtering and analysis; from this procedure, we can obtain measurements of the intensity of the scattered light changes. From experiments previously reported by our research group, we know that this detection method yields signal intensities consistently proportional to the concentration of suspended nanowires [8,9].

## 3. Results and Discussion

### 3.1. Structural Characterization

Figure 3 shows SEM imaging of NiNWs with average sizes of about 30 nm in diameter and 800 nm in length; due to these nanoparticles’ strong magnetic properties, they have a strong tendency to bundle when characterized in dry conditions.

Using bright-field (BF) and dark-field (DF) scanning/transmission electron microscopy (STEM), the morphology of gelatin-coated NiNWs and gelatin nanoparticles were determined, as shown in Figure 4. Figure 4a,b show gelatin-coated NiNW micrographs by BF- and DF-STEM, respectively; From Figure 4c,d, we can observe the incorporation of AgNPs into the gelatin-coated NiNWs, BF- and DF-STEM, respectively. It is noteworthy that DF-STEM images are associated with the atomic number (Z contrast); in this regard, it is possible to differentiate the elements that compose the sample and corroborate that NiNWs are coated with an AgNP/gelatin matrix. Figure 4e,f show that this methodology allows the detection of silver nanoparticles within gelatin nanoparticles.

### 3.2. Chemical Characterization

Gelatin-coated NiNWs with/without AgNPs were dispersed in 15 mL deionized water and labeled as NiNW-gelatin and NiNW-Ag-gelatin, respectively. ICP-OES analyzed these suspensions to determine their Ni content; Table 1 shows the results of the chemical analysis of gelatin-coated NiNW suspensions. This characterization aims to correlate the amount of NiNW in the suspension to the actual measurements made using our proposed magnetic alignment light-scattering technique.

### 3.3. Detection

Aliquots of 10 mL each of NiNW-gelatin, NiNW-Ag-gelatin, and as-synthesized non-coated NiNW suspension were prepared from known concentrations. These aliquots were tested in our magnetically responsive detection device to determine the detection capabilities of gelatin-coated nanowires. Before optical detection measurements were taken, all samples were ultrasonically agitated for 4 min at 30 W.

Detection measurements were performed by applying oscillating magnetic (on/off) fields at 6 Hz for an integration duration of 20 min; the signal-to-noise ratio was calculated using the difference of the signal change when no magnetic field was applied (baseline). A computer interface collected NiNWs’ optical dispersion data to perform statistical analysis such that the signal-to-noise ratio (SNR) could be computed. The SNR data of different nanoparticle dilutions are shown in Figure 5, and the results show that as the concentration of NiNWs decreases, the signal decreases. The NiNW-Ag-gelatin complexes exhibit higher SNR than NiNW-gelatin, especially at very low concentrations; however, the non-coated NiNWs show the highest signal intensity. From these measurements, it is remarkable that even at a very low concentration of 140 pg/mL, in the case of NiNWs coated in gelatin, we obtained a discernable signal of 13 dB. This result is promising in that the particles synthetized in this work can be detected at a relatively low threshold level. In comparison, other high-complexity techniques such as ICP-OES can typically detect Ni at concentrations as low as 3 ng/mL [25], and novel research related to Magneto resistance sensors reports the achievement of detecting magnetic nanoparticles as low as 10 ng/mL [17,26].

Colloidal stability tests were performed by measuring the signal provided for 4 h by the following colloidal suspensions: NiNW-gelatin, NiNW-Ag-gelatin, and non-coated NiNWs. The concentration of NiNWs was adjusted to yield an equivalent dispersion signal for the three samples. The results shown in Figure 6 suggest that the trend of signal decline between the three samples is very similar, so it can be inferred that a gelatin coating does not negatively affect nanowire stability. In these plots, we can notice the presence of steps during the measurement; we attribute these steps to the luminous noise present in the laboratory.

## 4. Conclusions

In this study, we have fabricated a core–shell composite with magnetic and optical properties that are able to be manipulated using an innovative yet straightforward magnetic alignment device for optical detection. Our composite is made of magnetic Ni nanowires coated with gelatin to provide a valuable surface for anchor proteins or nanoparticle decoration. The development of the procedure required extensive tests to find the optimal parameters to achieve the gelatin coating, avoiding aggregation, which is a common problem in the synthesis of nanoparticles. Furthermore, the addition of AgNPs demonstrates the capacity of the proposed encapsulation process to hold nanoparticles.

To test the proposed approach, we measured the concentration of Ni nanowires coated with gelatin. The measurements of nanocomposite concentration using our optical device correlate well with independent concentration measurements performed using an inductively coupled plasma optical emission spectrometer (ICP-OES). Furthermore, our results show that our magnetic nanocomposites are detectable at concentrations as low as 0.14 ppb; this high detectability suggests their potential in practical applications as labels to detect biological analytes. In this context, gelatin coating is a crucial aspect of our nanocomposite since it is an organic polymer—biocompatible with most proteins and other biomolecules—that can anchor antigens and antibodies in biosensing applications such as cancer detection.

## Figures and Tables

**Figure 1 nanomaterials-13-00015-f001:**
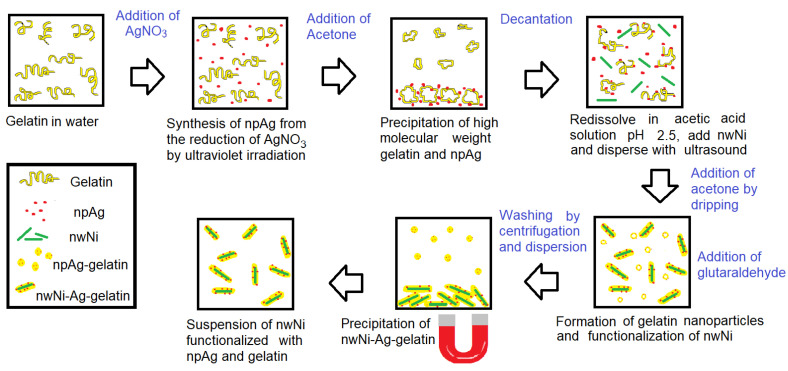
Schematic of the coating of NiNWs with AgNPs/gelatin.

**Figure 2 nanomaterials-13-00015-f002:**
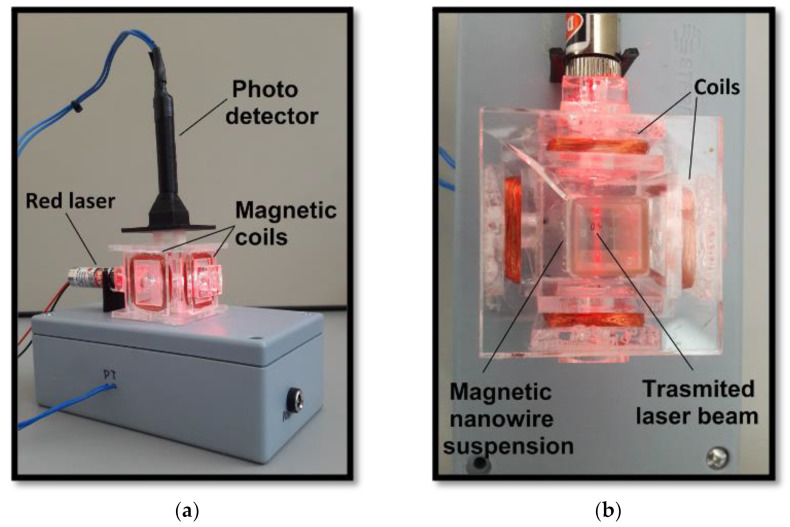
The experimental device designed to detect magnetic particles: (**a**) photo of the ensemble, (**b**) top view showing the laser light scattering caused by the NiNWs’ rotation.

**Figure 3 nanomaterials-13-00015-f003:**
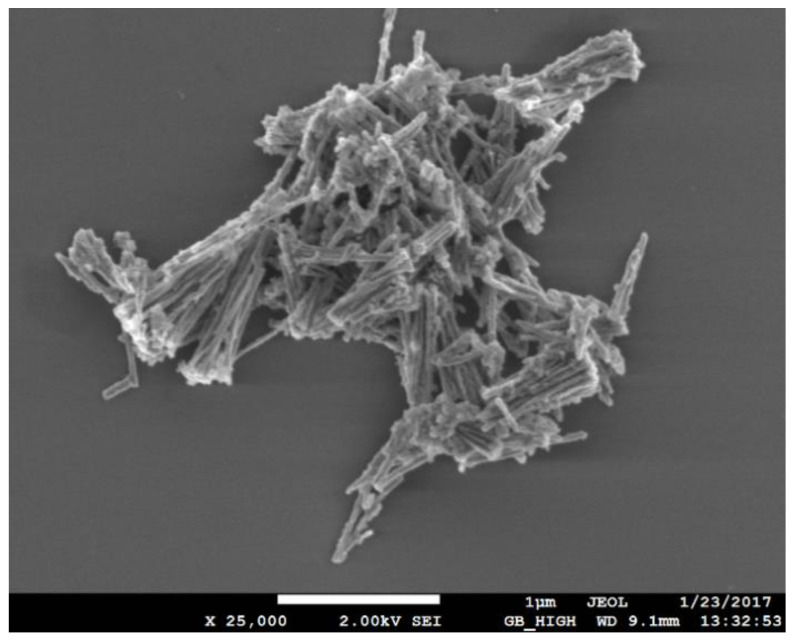
SEM micrograph of Ni nanowires.

**Figure 4 nanomaterials-13-00015-f004:**
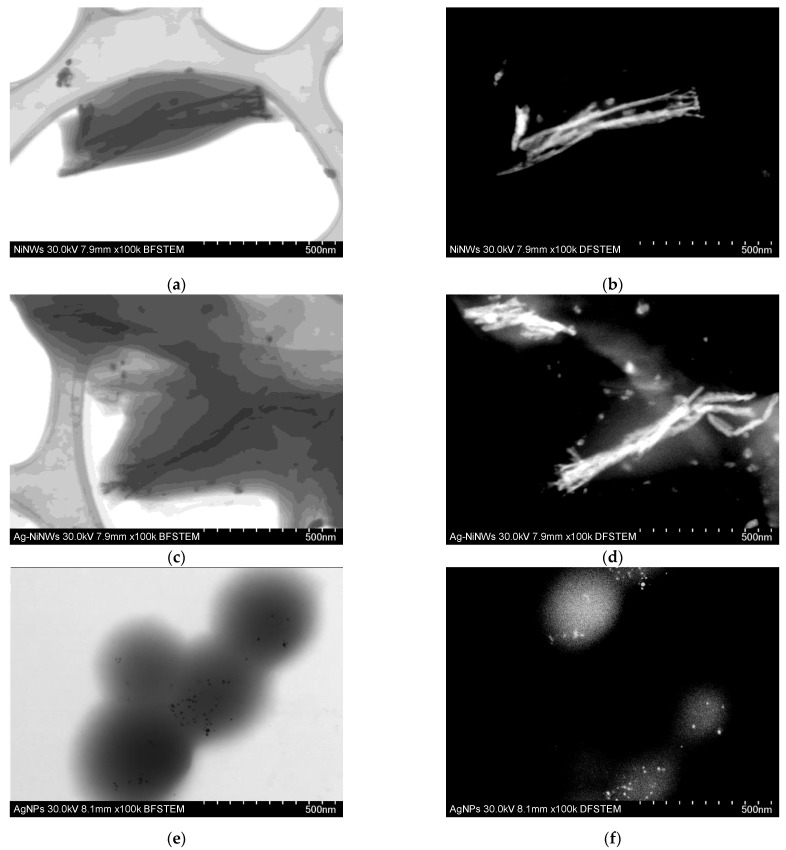
Bright-field (**right**) and dark-field (**left**) micrographs of (**a**,**b**) gelatin-covered NiNWs, (**c**,**d**) AgNP/gelatin-coated NiNWs, and (**e**,**f**) AgNP/gelatin nanoparticles decorated with silver nano particles to demonstrate the capacity to easily embed nanostructured components inside a gelatin matrix.

**Figure 5 nanomaterials-13-00015-f005:**
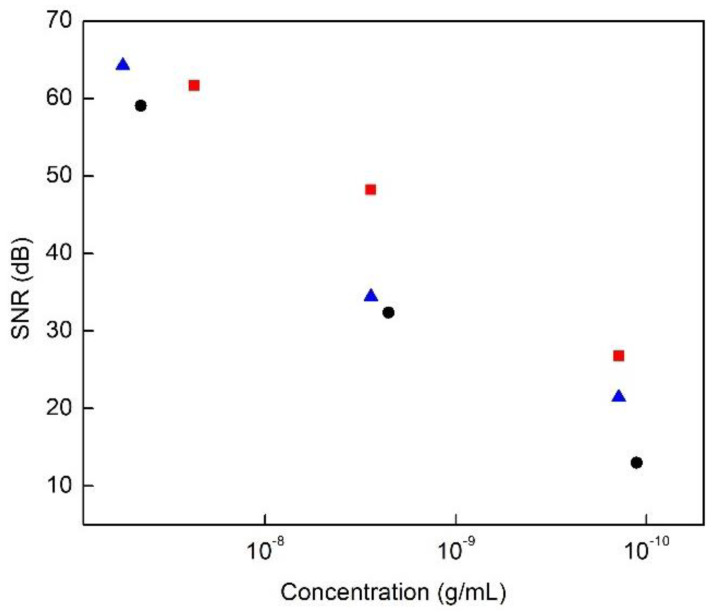
SNR of different concentrations of NiNWs: as-synthesized non-coated NiNWs (red-square), NiNW-Ag-gelatin (blue-triangle), and NiNW-gelatin (black-circle).

**Figure 6 nanomaterials-13-00015-f006:**
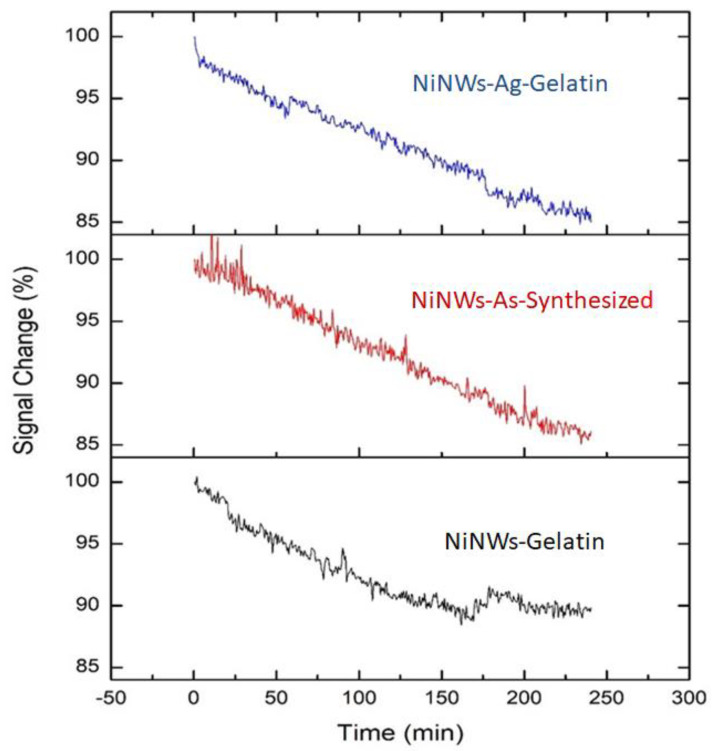
Signal change in time.

**Table 1 nanomaterials-13-00015-t001:** Results of ICP analyses for gelatin-coated nanowire samples.

Sample	ppb	Standard Deviation	g/mL
**NiNW-gelatin**	45	1.0	9.0 × 10^−7^
**NiNW-Ag-gelatin**	56	0.7	11.2 × 10^−7^

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
