# Peer review of "Gelatin-Coated Magnetic Nanowires for High-Sensitivity Optical Labels"

_nanomaterials, 2022, doi:10.3390/nano13010015_

Round 1
Reviewer 1 Report
In the present work, a novel optical detection label, gelatin-coated NiNWs, was reported. Compared with the traditional commercial ICP-OES and magnetoresistance sensing, the synthesized gelatin-coated particles have much more excellent detectability (~140 pg/mL), which makes it a good candidate for optical detection labels. I recommend publication after the follow comments are addressed.
Comments:
Q1: There are some grammatical errors, so the English should be improved in the revised manuscript.
Q2: The more detail Experimental procedure should be provided.-The authors mentioned that synchronizing on/off the magnetic fields in each direction promotes the rotation of the nanowires. This on/off procedure causes a change in the scattered light intensity of the rotating nanowires; this phenomenon is the key to detecting suspended nanowires. Does the nanowire return to its original state after the magnetic field is removed? How to ensure the repeatability and consistency of test results when using as-synthesized nanowire as a label?
Q3: The NiNWs showed in Fig. 3 are seriously agglomerated and need to be fully dispersed before characterization.
Q4: Fig. 4 (e) is not clear enough.
Q5: It is suggested that the detectability of the as-synthesized nanowires was presented.
Reviewer 2 Report
In the submitted manuscript, the authors have proposed the encapsulation of magnetic Ni nanowires with gelatin as an alternative for optical detection labels. Hence, they present a method for coating magnetic nanowires with gelatin and decorate them with Ag nanoparticles. The work is well suited for the journal and interesting enough for its publication but there are several issues to be addressed before I recommend its publication. In particular, my commentaries are the following ones:
a) There are several typos, grammar and/or style errors. Please, check the grammar carefully. I put here several examples.
i) Line 18: it should be “pg/ml” instead or “pg/mL”
ii) Line 41-2: I suggest “…detectability. Although more studies…, in this work, the encapsulation of the nanowire inside a gelatin matrix is proposed…”
iii) Lines 58: I suggest: “…changes when a magnetic field is applied!”
iv) Line 98 and 105: “coating” better than “coated”
v) Line 134: “ca.” (i.e., circa) is used only for time. Therefore I would put “roughly” or “about”
vi) line 167. I would recommend “when no magnetic field is applied” instead of the current phrase.
b b) Line 81: Here is not explained why the UV is needed (is explained in lines 100-101). I think that it should be explained also here (in that case, the explanation in following section could be shortened)
c c) Structural characterization: I think that the dark/bright field micrographs are fairly useful and adequate for the characterization. However, I was wondering if no measurements of EDX could also have been obtained (focusing on some of the regions) to confirm even more strongly the presence of Ag and/or Ni. A color map with EDX (with one color for each element) could have been shown. (Nevertheless, I think that the results with bright and dark field are good enough. I was wondering why not EDX measurements were also made)
d d) Detection section: In the abstract and in the conclusions is said that “the synthesized gelatin-coated particles have excellent detectability to the level of 140 pg/mL” but no discussion is made here about that. I think that it should also be emphasized in this section (and the comparison with other techniques should also be explained in more detail here -with references-)
e e) Figure 6 and related discussion: I think that it would be better to indicate directly in the figure the corresponding sample (for example, writing NiNWs-Ag-Gelatin in the upper figure and so on). Additionally, the trend of the signal decline is similar but in both cases with gelatin, there are several steps more marked than in the case without gelatin. I think that it should be at least commented and discussed a little bit.
Round 2
Reviewer 2 Report
The authors have answered properly to the questions of the reviewers and the manuscript has improved strongly (even though it was already quite well). Therefore, I recommend its publication.